# Effects of Non-Heated and Heat Processed Krill and Squid Meal-Based Diet on Growth Performance and Biochemical Composition in Juvenile Pacific Bluefin Tuna *Thunnus orientalis*

**Jeong-Hyeon Cho** [1,†] , **Takayuki Kurimoto** [1,‡] , **Yutaka Haga** [1,*] , **Yuji Kamimura** [2] , **Akira Itoh** [3] **and Shuichi Satoh** [1,§]

1 Graduate School of Marine Science and Technology, Tokyo University of Marine Science and Technology, Konan, Minato 4-5-7, Tokyo 108-8477, Japan; cjh0123@korea.kr (J.-H.C.); taka127coco@gmail.com (T.K.); ssatoh@fpu.ac.jp (S.S.)

2 Amami Fish Farm Co., Shinokawa Branch, Oshima, Kagoshima 894-1742, Japan; y-kamimura@maruha-nichiro.co.jp

3 Maruha Nichiro Co., Koto, Tokyo 135-8608, Japan; a-itoh@maruha-nichiro.co.jp

* Correspondence: haga@kaiyodai.ac.jp; Tel.: +81-3-5463-0555

† Current address: Jeju Fisheries Research Institute, National Institute of Fisheries Science, Jeju City 63610, Korea.

‡ Current address: The Nisshin OilliO Group, Ltd., Yokohama 235-8558, Japan.

§ Current address: Faculty of Marine Science and Technology, Fukui Prefectural University, Fukui 917-0003, Japan.

**Abstract:** This study investigated the effects of krill and squid meal and their heat processing on the growth performance and biochemical composition of juvenile Pacific bluefin tuna (PBT) *Thunnus orientalis*. An experiment using a 2 × 2 factorial design examined the effects of two dietary protein sources (squid and krill meal) and heat treatment (heated and non-heated). Prey fish were provided to a reference group. Fish with an initial mean weight of 74.1 mg were fed one of the five diets. After six days of the feeding trial, the fish fed with krill meal and non-heated diets showed improved growth compared to those fed with the squid meal and heated diets. Fish fed the non-heated diets showed significantly higher whole-body crude protein and crude lipid contents than fish fed the heated diets. These results suggest that nutrient availability could be improved by using krill meal and the non-heated treatment to improve the growth performance of juvenile PBT.

**Keywords:** Pacific bluefin tuna; squid meal; krill meal; heat treatment; growth performance

## 1. Introduction

Tuna aquaculture is one of the most important aquacultures worldwide. In 2017, the global production of tuna species was approximately 7.89 million tonnes [1]. Japan is not only one of the countries with the highest consumption of tuna species in the world, but total tuna production was as high as 195,200 metric tons in 2018 [2]. In particular, the total production of Pacific bluefin tuna (PBT) in Japan gradually increased from 2,000 metric tons in 2000 to 17,600 metric tons in 2018 [2,3]. This species is a top predator, grows quickly, and requires high dietary protein. Aquafeed is typically based on fish meal which is made from forage fish, such as sardine *Sardinops melanostictus* and anchovy *Engraulis ringens*; the major producers and exporters are Peru and Chile [4]. However, the rapid rise in global demand for aquafeed has reflected soaring prices [5]. In addition, fish meal production relies on natural fish resources. The dependence of the aquaculture industry on high dietary fish meal consumption is a serious concern regarding its sustainability. To overcome these limitations, considerable research efforts are being made to reduce the dependency of aquafeed manufacturers on fish meal [6]. In the last few decades, several studies have focused on plant proteins that can be utilized as fish meal alternatives [7–10]. However,

plant protein sources have relatively imbalanced amino acid profiles, low palatability, low nutrient digestibility, and anti-nutritional factors are present [6,9,11–15].

In terms of marine resources, Antarctic krill *Euphausia superba* is one of the most promising resources because of large biomass levels unparalleled anywhere else in the world's oceans. Furthermore, it has been commercially harvested since the 1960s, and today it is targeted by active fisheries of several nations [16,17]. Compared to conventional animal proteins, krill has several advantages such as similar amino acid and fatty acid composition with those of marine farmed fish [18,19] and higher contents of carotenoids, chitin, nucleotides, phospholipids, taurine, and vitamins [20–23]. Squid has been another popular aquafeed ingredient since the 1980s. Squid meal has high protein and strong palatability, is rich in taurine and polar lipids, and excellent feed ingredient [24–27]. Furthermore, improvement of egg quality and fish growth by squid meal based-diet were reported [24,28]. Although krill and squid meals are expensive as the main protein source due to their high unit price, they are ideal as supplemental proteins for low fishmeal diet that can effectively provide the nutrients that are lacking in plant ingredients.

Free amino acids and peptides are considered solubilized dietary proteins because they are water soluble and easily absorbed [29]. Most of the raw protein sources are denatured during heat cooking process, which reduces utility of amino acid and peptide from it. The krill and squid meals available in the market are heat processed, and therefore, they have low nutritive value because of decreased levels of free amino acids and peptides. Therefore, if squid and krill meal without heat processing are available, they may be an effective protein source for tuna. Previously, we showed that higher feed consumption and growth of red sea bream *Pagrus major* fed non-heated krill and squid meal-based diets [28]. However, there are few studies investigate utility of squid and krill meal without heat processing as dietary ingredients for tuna.

Therefore, this study aimed to evaluate the effect of krill and squid-based diets and their heat treatment on the biochemical composition and growth performance of PBT juveniles.

## 2. Materials and Methods

### 2.1. Experimental Diets

The test diets were formulated with 51.4% of four different animal protein sources (heated squid meal, HS; non-heated squid meal, NHS; heated krill meal, HK; non-heated krill meal, NHK). Raw squid and krill were pulverized using a centrifugal milling devise equipped with sieve (2.5 mm diameter, ZM500, Retsch Co, Clifton, NJ, USA). The resultant pulverized was kept under −30 °C. The frozen samples were then lyophilized by an automatic vacuum freeze drying devise (RLE-206II, Kyowa Vacuum Eng., Co, Saitama, Japan). The freeze-dried squid or krill meal were sieved using a 500 μm sieve. To make the HK and HS, the freeze-dried meals were kept for 12 h under 105 °C. Porcine blood meal, defatted horse mackerel meal, albumin from hen egg, DHA70E, fish oil, vitamin E, α-starch, taurine, calcium phosphate, choline chloride, sodium ascorbyl phosphate, mineral and vitamin premix, and bonito peptide were included in the test diets (Table 1). These ingredients were mixed and pelletized by fluidized bed granulation which is suspending particles in an air stream and liquid binder such as carboxy methyl cellulose is sprayed from the top of the system down onto the fluidized bed. The resultant pellets (ca. Ø 750 μm) were sieved. The moisture of diets (sinking pellets) was removed in the freeze dryer for 12 h and then kept under −30 °C until use. Spangled emperor *Lethrinus nebulosus* larvae which is popular prey for PBT larvae, was used as control (prey fish, PF). Fertilized eggs obtained from spangled emperor were introduced and hatched in 200 L aquaria.

**Table 1.** Formula, proximate composition, and water-soluble and -insoluble protein content in the test diets.

| Ingredients (%) [2] | Treatment [1] | | | | |
|---|---|---|---|---|---|
| | **HS** | **NHS** | **HK** | **NHK** | **PF** |
| Heated squid meal | 51.4 | - | - | - | - |
| Non-heated squid meal | - | 51.4 | - | - | - |
| Heated krill meal | - | - | 51.4 | - | - |
| Non-heated krill meal | - | - | - | 51.4 | - |
| Porcine blood meal | 5.0 | 5.0 | 5.0 | 5.0 | - |
| Defatted horse mackerel meal | 12.0 | 12.0 | 12.0 | 12.0 | - |
| Chicken egg albumin | 6.6 | 6.6 | 6.6 | 6.6 | |
| Chicken egg lecithin | 3.2 | 3.2 | 3.2 | 3.2 | - |
| DHA70E [3] | 1.3 | 1.3 | 1.3 | 1.3 | - |
| Taurine | 1.0 | 1.0 | 1.0 | 1.0 | - |
| Fish oil [4] | 6.7 | 6.7 | 8.1 | 8.1 | - |
| α-Starch | 2.0 | 2.0 | 2.0 | 2.0 | - |
| Monobasic calcium phosphate | 1.9 | 1.9 | 1.9 | 1.9 | - |
| Vitamin E (50%) | 0.1 | 0.1 | 0.1 | 0.1 | - |
| Choline chloride | 0.8 | 0.8 | 0.8 | 0.8 | - |
| Sodium ascorbyl phosphate | 0.1 | 0.1 | 0.1 | 0.1 | - |
| Mineral mixture [5] | 1.5 | 1.5 | 1.5 | 1.5 | - |
| Vitamin mixture [6] | 3.0 | 3.0 | 3.0 | 3.0 | - |
| Bonito peptide [7] | 2.0 | 2.0 | 2.0 | 2.0 | - |
| Carboxymethyl cellulose | 2.0 | 2.0 | 2.0 | 2.0 | - |
| Cellulose | 1.4 | 1.4 | - | - | - |
| Proximate composition (%, dry-weight) | | | | | |
| Moisture | 7.6 | 6.6 | 6.3 | 7.6 | 91.8 |
| Crude protein | 57.2 | 58.4 | 56.8 | 56.2 | 63.7 |
| Water-soluble protein | 4.1 | 7.8 | 3.3 | 4.9 | - |
| Water-insoluble protein | 48.9 | 48.6 | 51.5 | 47.6 | - |
| Crude lipid | 22.4 | 24.8 | 21.2 | 21.8 | 22.4 |
| Crude ash | 8.3 | 8.0 | 13.2 | 14.1 | 12.3 |

[1] HS: heated squid meal; NHS: non-heated squid meal; HK: heated krill meal; NHK: non-heated krill meal; PF: prey fish, Spangled emperor larvae *Lethrinus nebulosus*. [2] Squid meal (CP: 67.6%, CL: 13.5%); Krill meal (CP: 67.7%, CL: 10.7%); Porcine blood meal (CP: 71.9%, CL: 1.3%); Defatted horse mackerel meal (CP: 78.6%, CL: 2.9%); Egg albumin (CP: 82.1%, CL: 0.1%). [3] DHA70E (Harima Foods Co., Osaka, Japan). [4] Cod liver oil (Kanematsu Shintoa Foods Co., Tokyo, Japan). [5] Mineral mixture (mg/kg diet): Na (as NaCl) 197; Mg (as MgSO$_4$·7H$_2$O) 735; Fe (as FeC$_6$H$_5$O$_7$·5H$_2$O) 258; Zn (as ZnSO$_4$·7H$_2$O) 40; Mn (as MnSO$_4$·5H$_2$O) 18; Cu (as CuSO$_4$·5H$_2$O) 3.9; Al (as AlCl$_3$·6H$_2$O) 0.56; Co (as CoCl$_2$·6H$_2$O) 0.15; I (as KIO$_3$) 0.89; α-cellulose carrier. [6] Vitamin mixture (amount/kg diet): thiamine hydrochloride, 60 mg; riboflavin, 100 mg; pyridoxine hydrochloride, 40 mg; cyanocobalamin, 0.1 mg; ascorbic acid, 5000 mg; niacin, 400 mg; calcium pantothenate, 100 mg; inositol, 2000 mg; biotin, 6 mg; folic acid 15 mg; *p*-aminobenzoic acid, 50 mg; vitamin K$_3$, 50 mg; vitamin A acetate, 9000 IU; vitamin D3, 9000 IU. [7] Feeding stimulants. Values of proximate composition are presented as means of triplication.

Moisture content in the formulated diets ranged from 6.3–7.6%, and in the PF group moisture content was 91.8%. The formulated diets were isonitrogenous (57%) and isolipidic (22%) (Table 1). Crude protein and lipid levels in the control were 63.7 and 22.4%, respectively. The NHS and NHK diets contained higher water-soluble protein than the heated meal diets (the protein contents of NHS and NHK were 7.8% and 4.9% compared to 4.1% and 3.3% for HS and HK, respectively). Total and free amino acid content in the non-heated meal diets were greater than those of the heated meal diets (Tables 2 and 3). The most abundantly observed free amino acids in the krill-based diets (HK and NHK) were arginine and glycine, whereas the NHS diet contained a significant portion of proline. The control diet PF had the highest gross free amino acid level (5.69 g/100 g) compared to the other diets (2.55~4.57 g/100 g) (Table 3). Docosahexaenoic acid (DHA) level of essential fatty acids was highest in the PF group (25.0% of total fatty acid), followed by the squid-based diets (15.0~18.3%) and the krill-based diets (11.5~11.7%), whereas eicosapentaenoic acid (EPA) was highest in the krill-based diets (11.5~11.7%), followed by the squid-based diets (8.2~10.7%) and the PF group (6.3%) (Table 4).

**Table 2.** Total amino acid content of the test diets (g/100 g, dry-weight).

| | HS | NHS | HK | NHK | PF |
|---|---|---|---|---|---|
| Essential amino acids | | | | | |
| Arginine | 2.25 | 2.91 | 2.68 | 2.83 | 3.06 |
| Lysine | 2.83 | 3.12 | 2.67 | 3.55 | 4.77 |
| Histidine | 1.86 | 1.00 | 0.92 | 1.15 | 2.71 |
| Phenylalanine | 1.93 | 2.11 | 1.99 | 1.99 | 2.27 |
| Leucine | 3.19 | 3.32 | 3.20 | 3.42 | 4.08 |
| Isoleucine | 1.27 | 1.42 | 1.32 | 1.30 | 1.39 |
| Methionine | 0.97 | 1.25 | 1.06 | 1.16 | 1.61 |
| Valine | 1.48 | 1.74 | 1.62 | 1.53 | 1.82 |
| Threonine | 1.99 | 2.02 | 1.87 | 1.82 | 2.82 |
| Tryptophan | 0.41 | 0.47 | 0.37 | 0.61 | 0.55 |
| Non-essential amino acids | | | | | |
| Taurine | 1.73 | 2.02 | 1.64 | 1.86 | 0.76 |
| Alanine | 2.86 | 2.89 | 2.83 | 2.68 | 2.96 |
| Tyrosine | 1.44 | 1.57 | 1.56 | 1.59 | 2.09 |
| Cystine | 0.43 | 0.50 | 0.33 | 0.39 | N/D |
| Cystathionine | 0.13 | 0.07 | 0.09 | 0.12 | 0.07 |
| Glycine | 2.18 | 2.40 | 2.76 | 2.91 | 2.14 |
| Glutamic acid | 6.09 | 6.82 | 6.40 | 6.33 | 6.69 |
| Serine | 2.25 | 2.37 | 2.22 | 2.22 | 3.67 |
| Aspartic acid | 4.42 | 4.70 | 4.73 | 4.67 | 4.26 |
| Proline | 1.79 | 2.56 | 1.73 | 2.29 | 2.35 |
| Total | 41.49 | 45.25 | 42.00 | 44.42 | 50.09 |

Values are presented as means of triplication. HS, heated squid meal; NHS, non-heated squid meal; HK, heated krill meal; NHK, non-heated krill meal; PF, prey fish; Spangled emperor larvae, *Lethrinus nebulosus*. N/D: not detected (detection limit: 0.01 g/100 g, dry-weight).

**Table 3.** Free amino acid content of the test diets (g/100 g, dry-weight).

| | HS | NHS | HK | NHK | PF |
|---|---|---|---|---|---|
| Essential amino acids | | | | | |
| Arginine | 0.07 | 0.14 | 0.17 | 0.36 | 0.58 |
| Lysine | 0.05 | 0.12 | 0.06 | 0.22 | 0.65 |
| Histidine | 0.06 | 0.06 | 0.04 | 0.01 | 1.08 |
| Phenylalanine | 0.01 | 0.08 | 0.02 | 0.07 | 0.33 |
| Leucine | 0.03 | 0.19 | 0.04 | 0.14 | 0.57 |
| Isoleucine | 0.02 | 0.08 | 0.03 | 0.07 | 0.16 |
| Methionine | N/D | 0.03 | N/D | 0.03 | 0.19 |
| Valine | 0.03 | 0.09 | 0.04 | 0.10 | 0.20 |
| Threonine | 0.03 | 0.08 | 0.03 | 0.06 | 0.13 |
| Tryptophan | N/D | 0.01 | N/D | 0.01 | 0.13 |
| Non-essential amino acids | | | | | |
| Taurine | 1.29 | 1.62 | 1.20 | 1.46 | 0.53 |
| Alanine | 0.15 | 0.28 | 0.11 | 0.24 | 0.31 |
| Tyrosine | 0.02 | 0.10 | 0.02 | 0.08 | 0.27 |
| Cystine | 0.03 | 0.04 | N/D | N/D | N/D |
| Cystathionine | N/D | 0.01 | 0.01 | 0.01 | 0.04 |
| Glycine | 0.04 | 0.11 | 0.48 | 0.74 | 0.08 |
| Glutamic acid | 0.01 | 0.11 | 0.01 | 0.03 | 0.21 |
| Serine | 0.03 | 0.07 | 0.03 | 0.05 | 0.09 |
| Aspartic acid | 0.06 | 0.11 | 0.06 | 0.05 | 0.10 |
| Proline | 0.62 | 1.23 | 0.42 | 0.58 | 0.04 |
| Total | 2.55 | 4.57 | 2.76 | 4.31 | 5.69 |

Values are presented as means of triplication. N/D: not detected (detection limit: 0.01 g/100 g, dry-weight). HS, heated squid meal; NHS, non-heated squid meal; HK, heated krill meal; NHK, non-heated krill meal; PF, prey fish; Spangled emperor larvae, *Lethrinus nebulosus*.

**Table 4.** Fatty acid composition (area% of total lipid) of the test diets.

| Fatty Acids | Experimental Diets | | | | |
|---|---|---|---|---|---|
| | HS | NHS | HK | NHK | PF |
| 14:0 | 3.7 | 3.8 | 5.5 | 6.2 | 4.4 |
| 16:0 | 18.0 | 17.5 | 19.2 | 17.5 | 19.7 |
| 16:1n−7 | 5.5 | 5.0 | 5.3 | 6.0 | 4.2 |
| 18:0 | 4.6 | 4.1 | 3.4 | 2.9 | 5.2 |
| 18:1n−9 | 11.7 | 10.3 | 12.0 | 11.3 | 9.7 |
| 18:1n−7 | 3.5 | 3.3 | 3.7 | 3.8 | 2.0 |
| 18:2n−6 | 2.8 | 2.2 | 4.1 | 3.4 | 2.7 |
| 18:3n−3 | 0.4 | 0.5 | 0.5 | 0.5 | 0.7 |
| 18:4n−3 | 1.0 | 1.0 | 1.4 | 1.4 | 1.5 |
| 20:1n−9 | 3.2 | 2.8 | 1.7 | 1.9 | 0.5 |
| 20:1n−11 | 2.7 | 2.5 | 1.4 | 1.6 | 0.3 |
| 20:4n−6 | 1.5 | 1.6 | 1.5 | 1.3 | 1.7 |
| 20:4n−3 | 0.4 | 0.4 | 0.3 | 0.3 | 0.5 |
| 20:5n−3 | 8.2 | 10.7 | 11.7 | 11.5 | 6.3 |
| 22:1n−11 | 4.0 | 3.1 | 2.8 | 2.8 | 0.3 |
| 22:1n−13 | 0.7 | 0.6 | 0.7 | 0.6 | 0.1 |
| 22:5n−3 | 0.9 | 0.8 | 0.5 | 0.5 | 1.1 |
| 22:6n−3 | 15.0 | 18.3 | 11.5 | 11.7 | 25.0 |
| 22:6n−3/20:5n−3 | 1.8 | 1.7 | 1.0 | 1.0 | 4.0 |
| Others | 12.1 | 11.5 | 12.8 | 15.0 | 13.6 |
| Σn−3 LC−PUFA [1] | 25.5 | 31.2 | 25.4 | 25.4 | 34.4 |

Values are presented as means of triplication. HS, heated squid meal; NHS, non-heated squid meal; HK, heated krill meal; NHK, non-heated krill meal; PF, prey fish; Spangled emperor larvae, *Lethrinus nebulosus*. [1] LC−PUFA, long chain poly unsaturated fatty acids; Σn−3 LC−PUFA: 18:4n−3, 20:4n−3, 20:5n−3, 22:5n−3, 22:6n−3.

*2.2. Feeding Experiment and Sampling Schedule*

Fertilized eggs were collected natural spawning of Pacific bluefin tuna (PBT) reared at Amami Fish Farm Co, Kagoshima, Japan. The rotifers *Brachionus rotundiformis* and *Artemia* nauplii were offered to larval PBT before they became 10 mm of total length prior to start of the feeding experiment. The larval PBT were fed larvae of spangled emperor *Lethrinus nebulosus* with yolk-sac until the PBT reached 20 mm in total length. Pacific bluefin tuna larvae with total length of 20.5 ± 0.2 mm (19 days post hatching, dph) were introduced into ten 500-L circular polycarbonate aquaria. The aquaria had walls carrying black tape in a checkered pattern to prevent collision of fish against aquarium walls [30] (Table 5).

**Table 5.** Rearing conditions for feeding trial of Pacific bluefin tuna juveniles.

| | |
|---|---|
| Average Total Length of Initial Fish (mm) [1] | 20.5 ± 0.2 |
| Average body weight of initial fish (mg) [1] | 74.1 ± 2.7 |
| Age of initial fish body (day post hatching) | 19 |
| Tank volume (L) | 500 |
| Number of fish (ind./tank) | 240 |
| Rearing period (days) | Weaning period 3 |
| | Sole feeding of test diet period 6 |
| Water temperature (°C) [2] | 27.8 ± 0.6 |
| pH [2] | 8.1 ± 0.0 |
| Dissolved oxygen (mg/L) [2] | 11.0 ± 1.7 |
| Photoperiod | 11L (07:00–18:00):13D |
| Exchange rate of sea water (% tank volume/day) | 1200 |
| Aeration (mL/min) | 800 |

[1] Mean ± standard deviation (SD) ($n = 100$). [2] Mean ± SD ($n = 10$).

The initial three days were designated as acclimation period for the formulated diet (19–21 dph), and the following six days were allocated for feeding with only the test diets for the juveniles (22–28 dph). A commercial diet for marine fish (CP: 58.1%, CL: 19.4%, dry

weight) was provided to all treatment groups hourly between 7:00 and 19:00 during the weaning period. Moreover, PF was offered to all groups (three times a day at 7:00, 13:00, and 19:00). The PF offered was reduced continuously; PF: PBT larvae (ind.: ind.) = 150:1, 120:1, and 90:1, so as to promote acclimation of PBT larvae to the experimental diets. PBT juveniles were fed only one of the experimental diets after a three-day weaning period. Fish in the PF group were subjected to satiation feeding of PF for whole period of the feeding trial. Dietary treatment was carried out in duplicate. Rearing of fish was finished after the sixth day from the beginning of the trial to have enough fish for chemical composition analysis. The daily seawater exchange rate was 1200% of the capacity of an aquarium. The average of dissolved oxygen concentration and water temperature were $11.0 \pm 1.7$ mg/L and $27.8 \pm 0.6$ °C, respectively (Table 5).

Prior to each sampling, fish were starved for 12 h. In this case, 50 fish were collected just before the beginning of the feeding trial and kept at $-80$ °C. PBT were also collected at 4, 8, and 10 days after feeding to evaluate fish growth. Ten fish were measured at four and eight days, and 120 fish were measured at ten days, for fork and total length. The whole body was weighed to obtain wet weight for calculation of the condition factor (CF). The carcasses were kept under $-80$ °C.

The bottom of aquaria was cleaned every day with a syphon, and survival of fish was estimated by counting the number of dead fish. Number of dead fish was subtracted from the fish number introduced at the start of the feeding trial for calculating survival rate. CF, specific growth rate (SGR), thermal growth coefficient (TGC), and weight gain (WG) were calculated by the formulae:

$$\text{WG (\%)} = [\text{wet weight of the final fish (g)} - \text{wet weight of the initial fish (g)}]/\text{wet weight of the initial fish (g)} \times 100$$

$$\text{SGR (\%)} = [\ln \text{wet weight of the final fish (g)} - \ln \text{wet weight of the initial fish (g)}] \times 100/\text{time (days)}$$

$$\text{TGC} = (\text{wet weight of the final fish (g)}^{1/3} - \text{wet weight of the initial fish (g)}^{1/3}) \times (\text{sum day-degrees Celsius})^{-1} \times 1000$$

$$\text{CF} = \text{wet weight (g)}/[\text{fork length (cm)}]^3 \times 100$$

### 2.3. Chemical Analyses

Diets and carcass were analyzed using standard methods for dry and wet matter, crude protein, and ash [31]. Chemical analysis was conducted in triplicate for each sample at the Laboratory of Fish Nutrition, Tokyo University of Marine Science and Technology, Minato, Tokyo, Japan, and the data was averaged. Moisture content was determined gravimetrically by drying the sample in a dry oven at 105 °C until achieving a constant weight. The samples were incinerated under 650 °C for 8 h by a muffle furnace (FO200, Yamato Co., Tokyo, Japan) for ash content determination. The Kjeldahl method with using an automatic titlator (Kjeltec 2400, FOSS Japan Co, Tokyo, Japan) and conversion index of 6.25 were employed for crude protein analysis. Dietary water-soluble protein content was determined according to de Schrijyer and Ollevier (2000) [32]. Chloroform and methanol mixture (2:1) was used for crude lipids extraction [33]. Preparation of fatty acid methyl ester (FAME) was followed the previously reported methods [34]. FAME was analyzed by a gas chromatograph (GC-2025, Shimadzu, Tokyo, Japan) installed with a fiberglass capillary column (30 m × 0.32 mm, i. d., SUPERCO-WAX10, Sigma-Aldrich Co, St. Louis, MO, USA) and peak area of FAME was measured by a recorder (C-R8A Chromatopac; Shimadzu). Temperature of the column oven was elevated from 170 °C at 2 °C/min for 40 min and kept under 250 °C. An automatic amino acid analyzer (JLC-500/v; JEOL Co., Tokyo, Japan) was used for the total and free amino acid analysis as previously described [35]. Samples were hydrolyzed with 10 mL 2% sulfosalicylic acid (*w/v*, Wako Fujifilm Co, Tokyo, Japan), or 4M methanesulfonic acid (Sigma-Aldridge Co, St. Louis, MO, USA) for free and total amino acid analysis, respectively. The digesta was homogenized for 1.5 min and centrifuged at $1610 \times g$ under 4 °C for 15 min twice by a devise (SRX-201; Tommy Co., Tokyo, Japan); the upper layers were pooled and filtered with a membrane filter (0.45 μm, Millipore Co,

Darmstadt, Germany). The filtrate was analyzed for amino acid. Constitutional amino acid content was calculated by subtracting the free amino acid from the total amino acid.

### 2.4. Statistical Analyses

Data on fish growth, initial and final carcass composition were analyzed by a one-way analysis of variance (ANOVA) followed by Tukey's multiple range tests. The main effects of dietary protein source and the heat processing were tested using a two-way ANOVA. Statistical significance was accepted when probability was below 95%. IBM SPSS 19 (SPSS Inc., Chicago, IL, USA) was used for statistical analysis.

## 3. Results

### 3.1. Survival and Growth of PBT Juveniles

The survival of juvenile PBT fed the treatment and control diets during the test period is presented in Figure 1. There were no significant differences in survival during the period from one to five days (including the weaning period of one to three days) in all treatment groups. However, significant differences in survival were observed thereafter until the final day. The highest survival of the final fish was found in the PF group (75.3%), followed by the NHK, HK, NHS, and HS groups (52.6%, 51.9%, 45.9%, and 25.8%, respectively). There were no significant differences among the HK, NHK, and NHS groups on the final day, but these groups differed significantly from the HS group. Survival in the krill-based diet groups (HK and NHK) was similar during the rearing period (Figure 1). Survival was not affected by the heat treatment. Effects were significant for protein sources during the rearing period; however, no interactive effects of two factors were detected (Table 6).

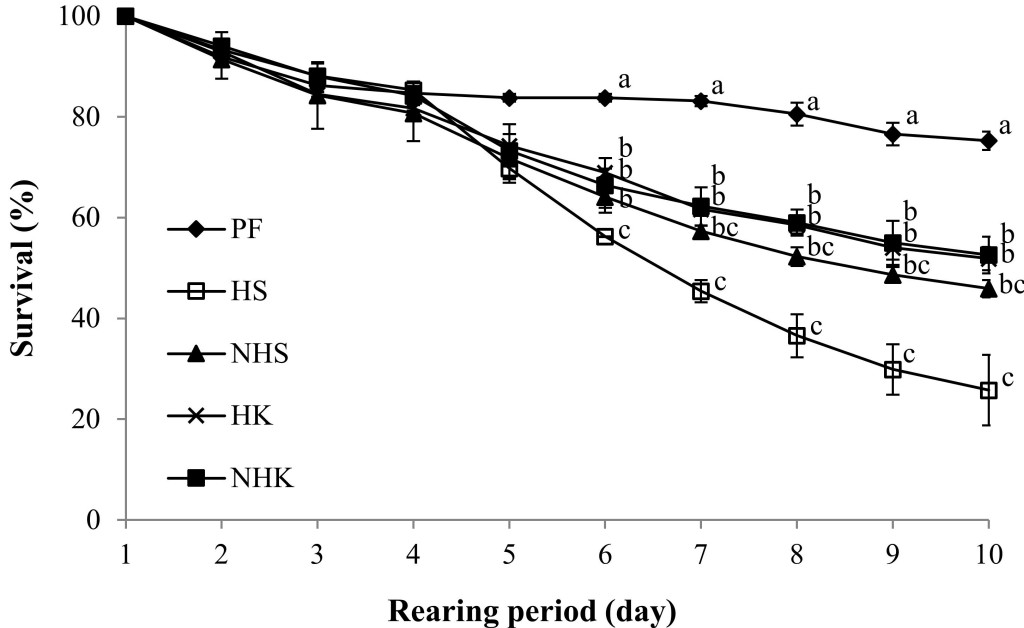

**Figure 1.** Survival of Pacific bluefin tuna juveniles fed different test diets during the rearing period. Different superscript letters indicate significant difference among the dietary groups (Tukey's test, *p* < 0.05). HS, heated squid meal; NHS, non-heated squid meal; HK, heated krill meal; NHK, non-heated krill meal; PF, prey fish; Spangled emperor larvae, *Lethrinus nebulosus*.

**Table 6.** Growth performance, proximate composition and biological indices of Pacific bluefin tuna fed the test diets [1] for 9 days.

| | Initial | Final (28 dph) | | | | | p | | |
|---|---|---|---|---|---|---|---|---|---|
| | (19 dph) | HS | NHS | HK | NHK | PF | I | H | I × H |
| | | | | Growth performance | | | | | |
| Total length (mm) | 20.5 ± 0.2 | 33.2 ± 0.2 [d] | 35.7 ± 0.3 [c] | 34.6 ± 0.3 [c] | 39.0 ± 0.3 [b] | 47.7 ± 0.4 [a] | * | * | * |
| Body depth (mm) | 4.2 ± 0.0 | 6.4 ± 0.0 [e] | 7.1 ± 0.1 [c] | 6.7 ± 0.1 [d] | 7.4 ± 0.1 [b] | 8.6 ± 0.1 [a] | * | * | ns |
| Body weight (mg) | 74.1 ± 2.7 | 251.0 ± 5.6 [d] | 355.3 ± 10.7 [c] | 318.6 ± 8.3 [c] | 480.4 ± 12.1 [b] | 837.4 ± 21.3 [a] | * | * | * |
| CF | 0.8 ± 0.1 | 0.7 ± 0.1 [c] | 0.8 ± 0.1 [b] | 0.7 ± 0.1 [b] | 0.8 ± 0.1 [a] | 0.8 ± 0.1 [b] | * | * | * |
| WG (%) | | 238.7 ± 3.1 [c] | 379.6 ±16.1 [c] | 330.0 ± 20.9 [c] | 548.3 ± 3.1 [b] | 1030.1 ± 76.4 [a] | * | * | ns |
| SGR (%) | | 13.6 ± 0.1 [d] | 17.5 ± 0.0 [c] | 16.2 ± 0.8 [c] | 20.5 ± 0.2 [b] | 27.4 ± 0.5 [a] | * | * | ns |
| TGC | | 0.2 ± 0.0 [d] | 0.4 ± 0.0 [c] | 0.3 ± 0.0 [cd] | 0.5 ± 0.0 [b] | 1.1 ± 0.1 [a] | * | * | ns |
| Survival rate (%) | | 25.8 ± 9.9 [c] | 45.9 ± 2.4 [bc] | 51.9 ± 3.3 [b] | 52.6 ± 5.2 [b] | 75.3 ± 2.5 [a] | * | ns | ns |
| | | | | Proximate composition (%, wet-weight) | | | | | |
| Moisture | 83.5 ± 0.2 | 83.7 ± 0.2 [ab] | 82.9 ± 0.1 [c] | 84.1 ± 0.3 [a] | 83.3 ± 0.0 [bc] | 83.4 ± 0.2 [bc] | * | * | ns |
| Crude protein | 12.4 ± 0.0 | 11.2 ± 0.3 [c] | 11.9 ± 0.1 [b] | 11.6 ± 0.1 [bc] | 12.6 ± 0.0 [a] | 12.9 ± 0.1 [a] | * | * | ns |
| Crude lipid | 2.0 ± 0.1 | 2.0 ± 0.0 [c] | 2.6 ± 0.1 [a] | 1.8 ± 0.1 [c] | 2.2 ± 0.0 [b] | 1.6 ± 0.0 [d] | * | * | * |
| Crude ash | 2.6 ± 0.1 | 3.2 ± 0.2 | 2.7 ± 0.2 | 2.9 ± 0.2 | 2.7 ± 0.1 | 2.8 ± 0.1 | ns | * | ns |

Values of total length, body depth, body weight and CF (condition factor) are means ± SD of 100 (19 dph) or 120 (28 dph) fish. Values are means ± SD of 2 groups of fish ($n = 2$; WG, SGR, TGC and survival rate) or 3 groups of fish ($n = 3$; proximate composition). Values in a same row with different letter are significantly different (Tukey's test, $p < 0.05$). I, ingredients (squid meal and krill meal); H, heated and non-heated treatment; ns, no significant difference (two-way ANOVA, $p > 0.05$); *, $p < 0.05$. WG, weight gain; SGR, specific growth rate; TGC, thermal growth coefficient. [1] HS, heated squid meal; NHS, non-heated squid meal; HK, heated krill meal; NHK, non-heated krill meal; PF, prey fish; Spangled emperor larvae, *Lethrinus nebulosus*.

The growth performances of juvenile PBT during all test periods are shown in Table 6, Figures 2–4. The best growth was found in the PF group, and its difference was significant among all groups ($p < 0.05$). The PBT juveniles in non-heated diet groups grew significantly better than those in the heated treatment diet groups ($p < 0.05$), and the fish in krill meal-based diet groups grew significantly better than those in the squid meal-based diet groups ($p < 0.05$) (Figures 2 and 3). The WG, SGR, and TGC of fish were affected by both protein sources and heat treatments, but there were no interactive effects of either factor (Figure 4).

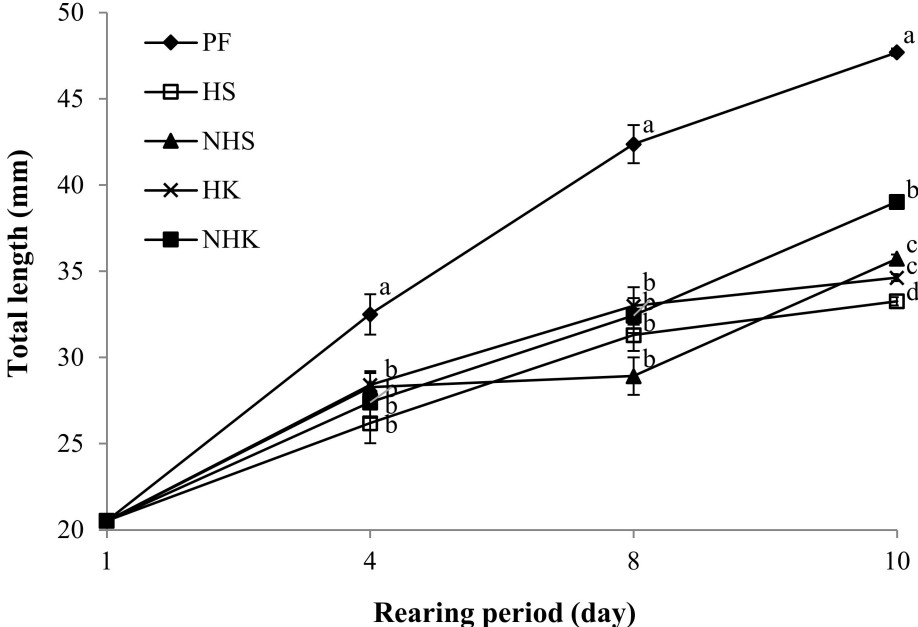

**Figure 2.** Final total length of Pacific bluefin tuna juveniles fed different test diets during the rearing period. Different superscript letters indicate significant difference among the dietary groups (Tukey's test, $p < 0.05$; 1st day, $n = 50$; 4th and 8th day, $n = 10$; 10th day, $n = 120$). HS, heated squid meal; NHS, non-heated squid meal; HK, heated krill meal; NHK, non-heated krill meal; PF, prey fish; Spangled emperor larvae, *Lethrinus nebulosus*.

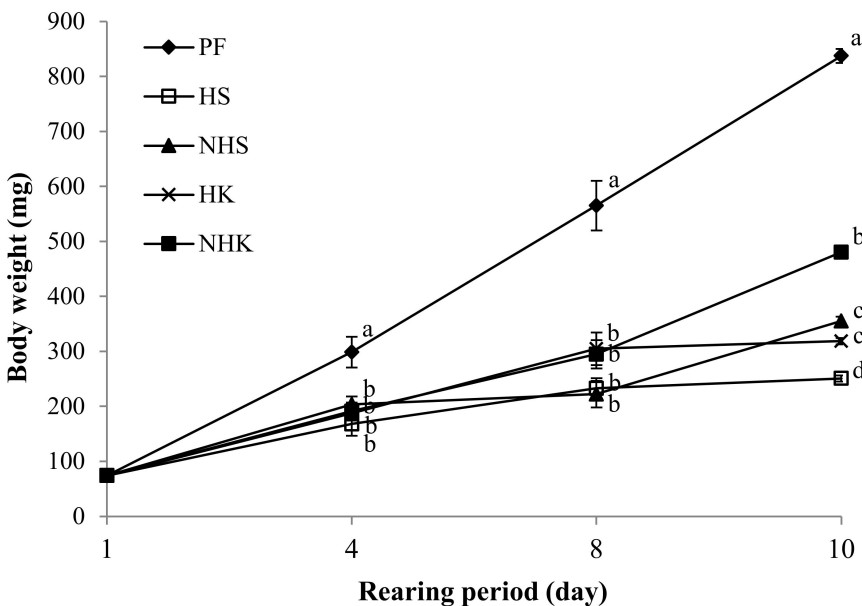

**Figure 3.** Final body weight of Pacific bluefin tuna juveniles fed different diets during the rearing period. Different superscript letters indicate significant difference among the dietary groups (Tukey's test, $p < 0.05$; 1st day, $n = 50$; 4th and 8th day, $n = 10$; 10th day, $n = 120$). HS, heated squid meal; NHS, non-heated squid meal; HK, heated krill meal; NHK, non-heated krill meal; PF, prey fish; Spangled emperor larvae, *Lethrinus nebulosus*.

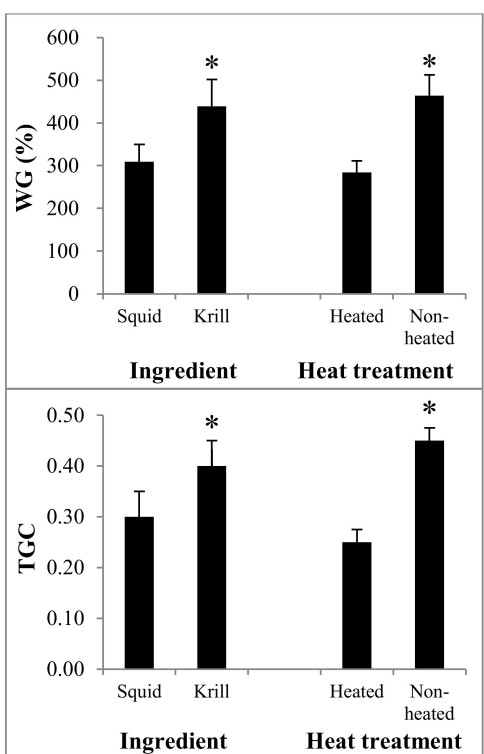

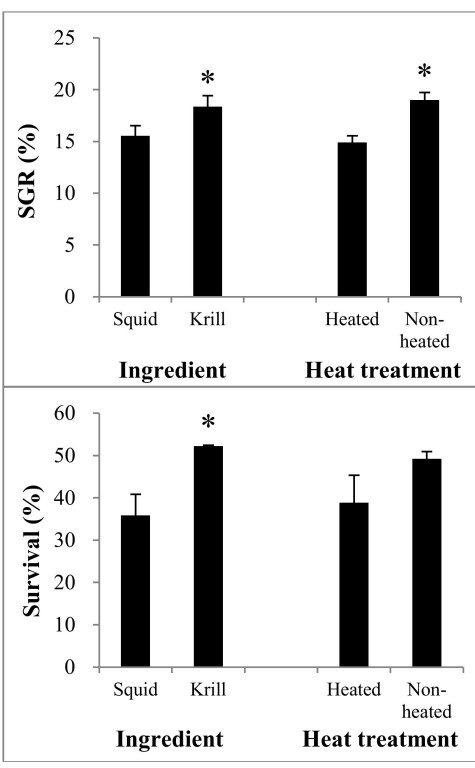

**Figure 4.** Means of WG (g, **upper-left**), SGR (%, **upper-right**), TGC (**lower-left**), and survival (%, **lower-right**) of juvenile Pacific bluefin tuna after rearing period for the main effects of ingredient and heat treatment. Data was expressed as means ± SE. WG, weight gain; SGR, specific growth rate; TGC, thermal growth coefficient. The asterisks indicate significant differences by two-way ANOVA analysis ($p < 0.05$).

### 3.2. Chemical Property of PBT

The highest moisture content was noted in the HK group, followed by the HS, PF, NHK, and NHS groups ($p < 0.05$). The crude protein contents in the PF and NHK groups were significantly higher than those in the other groups, whereas the crude lipid content in the NHS group was significantly higher than in the other groups ($p < 0.05$). No significant differences were observed in the crude ash among all the groups ($p > 0.05$). The moisture, crude protein, and crude lipid of the carcasses were affected by both protein source and heat treatment, but an interactive effect between both factors was detected only for crude lipids. By the final day, the CF observed in the NHK group was significantly higher than those of the PF, NHS, and HK groups, which were significantly higher than that of the HS group ($p < 0.05$).

Significant differences between the PF and the other groups were observed in all constitutional amino acid levels except valine and tryptophan. Specifically, most of the amino acid levels were higher in the PF group than in the other treatment groups. Constitutional amino acids were affected by both the protein source and heat treatment, while the free amino acids tended to be affected by the interaction. Only free histidine was affected by both factors (Table 7).

**Table 7.** Constitutional and free amino acid content of Pacific bluefin tuna fed the test diets [1] for 9 days (g/100 g, dry-weight).

| | Constitutional Amino Acid | | | | | | | | |
|---|---|---|---|---|---|---|---|---|---|
| | Initial | Final (28 dph) | | | | | *p* | | |
| | (19 dph) | HS | NHS | HK | NHK | PF | I | H | I × H |
| **Essential amino acid** | | | | | | | | | |
| Arginine | 2.56 ± 0.27 | 1.67 ± 0.47 [c] | 2.31 ± 0.16 [bc] | 2.53 ± 0.16 [ab] | 2.65 ± 0.09 [ab] | 3.14 ± 0.07 [a] | * | * | ns |
| Lysine | 3.36 ± 0.38 | 2.21 ± 0.48 [c] | 3.06 ± 0.22 [b] | 3.30 ± 0.11 b | 3.52 ± 0.08 [ab] | 4.14 ± 0.04 [a] | * | * | ns |
| Histidine | 0.77 ± 0.15 | 0.53 ± 0.15 [c] | 0.76 ± 0.04 [b] | 0.82 ± 0.03 [ab] | 0.88 ± 0.02 [ab] | 1.01 ± 0.03 [a] | * | * | ns |
| Phenylalanine | 1.89 ± 0.25 | 1.42 ± 0.37 [b] | 1.84 ± 0.13 [ab] | 2.05 ± 0.04 [a] | 2.05 ± 0.10 [a] | 2.34 ± 0.07 [a] | * | * | ns |
| Leucine | 3.18 ± 0.40 | 2.35 ± 0.50 [c] | 3.08 ± 0.24 [b] | 3.34 ± 0.11 [ab] | 3.48 ± 0.09 [ab] | 3.82 ± 0.11 [a] | * | * | ns |
| Isoleucine | 1.25 ± 0.16 | 0.93 ± 0.19 [b] | 1.11 ± 0.21 [ab] | 1.21 ± 0.10 [ab] | 1.31 ± 0.05 [a] | 1.45 ± 0.03 [a] | * | ns | ns |
| Methionine | 0.92 ± 0.12 | 0.84 ± 0.29 [b] | 0.93 ± 0.14 [b] | 1.21 ± 0.08 [ab] | 1.18 ± 0.13 [ab] | 1.45 ± 0.10 [a] | * | ns | ns |
| Valine | 1.28 ± 0.18 | 0.98 ± 0.49 | 1.14 ± 0.20 | 1.28 ± 0.11 | 1.31 ± 0.03 | 1.53 ± 0.04 | ns | ns | ns |
| Threonine | 2.24 ± 0.25 | 1.66 ± 0.28 [b] | 2.14 ± 0.08 [a] | 2.21 ± 0.04 [a] | 2.39 ± 0.08 [a] | 2.50 ± 0.04 [a] | * | * | ns |
| Tryptophan | 0.39 ± 0.08 | 0.26 ± 0.13 | 0.41 ± 0.05 | 0.47 ± 0.07 | 0.36 ± 0.06 | 0.50 ± 0.11 | ns | ns | * |
| | Free amino acid | | | | | | *p* | | |
| **Essential amino acid** | | | | | | | | | |
| Arginine | 0.39 ± 0.04 | 0.79 ± 0.20 [a] | 0.35 ± 0.25 [bc] | 0.29 ± 0.01 [bc] | 0.57 ± 0.02 [ab] | 0.15 ± 0.00 [c] | ns | ns | * |
| Lysine | 0.49 ± 0.01 | 0.84 ± 0.16 [a] | 0.43 ± 0.26 [bc] | 0.36 ± 0.01 [bc] | 0.61 ± 0.02 [ab] | 0.18 ± 0.00 [c] | ns | ns | * |
| Histidine | 0.69 ± 0.01 | 0.29 ± 0.04 [d] | 0.43 ± 0.05 [c] | 0.33 ± 0.00 [d] | 0.52 ± 0.01 [b] | 0.78 ± 0.01 [a] | * | * | ns |
| Phenylalanine | 0.32 ± 0.00 | 0.50 ± 0.16 [a] | 0.21 ± 0.17 [b] | 0.17 ± 0.01 [b] | 0.38 ± 0.01 [ab] | 0.11 ± 0.00 [b] | ns | ns | * |
| Leucine | 0.46 ± 0.01 | 0.75 ± 0.22 [a] | 0.31 ± 0.26 [b] | 0.26 ± 0.01 [b] | 0.52 ± 0.02 [ab] | 0.17 ± 0.00 [b] | ns | ns | * |
| Isoleucine | 0.23 ± 0.00 | 0.35 ± 0.09 [a] | 0.16 ± 0.13 [b] | 0.13 ± 0.00 [b] | 0.26 ± 0.01 [ab] | 0.10 ± 0.00 [b] | ns | ns | * |
| Methionine | 0.22 ± 0.00 | 0.32 ± 0.09 [a] | 0.15 ± 0.11 [ab] | 0.13 ± 0.00 [b] | 0.25 ± 0.01 [ab] | 0.08 ± 0.01 [b] | ns | ns | * |
| Valine | 0.37 ± 0.01 | 0.52 ± 0.12 [a] | 0.23 ± 0.19 [b] | 0.19 ± 0.00 [b] | 0.36 ± 0.01 [ab] | 0.17 ± 0.00 [b] | ns | ns | * |
| Threonine | 0.25 ± 0.00 | 0.35 ± 0.07 [a] | 0.19 ± 0.12 [b] | 0.15 ± 0.00 [b] | 0.25 ± 0.01 [ab] | 0.14 ± 0.00 [b] | ns | ns | * |
| Tryptophan | 0.05 ± 0.02 | 0.15 ± 0.07 [a] | 0.04 ± 0.04 [b] | 0.04 ± 0.00 [b] | 0.09 ± 0.01 [ab] | 0.02 ± 0.00 [b] | ns | ns | * |

Values are presented as means of triplication. Values in a same row with different superscript letters are significantly different (Tukey's test, $p < 0.05$). I, ingredients (squid meal and krill meal); H, heated and non-heated treatment; ns, no significant difference (two-way ANOVA, $p > 0.05$); *, $p < 0.05$. [1] HS, heated squid meal; NHS, non-heated squid meal; HK, heated krill meal; NHK, non-heated krill meal; PF, prey fish; Spangled emperor larvae, *Lethrinus nebulosus*.

The major fatty acids in the carcass were 16:0 (palmitic), 18:0 (stearic), 18:1n−9 (oleic), 20:5n−3 (eicosapentaenoic, EPA), and 22:6n−3 (docosahexaenoic, DHA) acids (Table 8). The sum of these major fatty acids accounted for more than 66.2%, 70.0%, 69.9%, 68.4%, and 77.4% of the total fatty acids in the HS, NHS, HK, NHS, and PF treatment groups, respectively. The major saturated fatty acid (SAFA) was 16:0 in all groups. Regarding polyunsaturated fatty acids (PUFAs), bluefin tuna is considered to be a good source of n−3 fatty acids, particularly DHA, which exhibited the highest levels in the PF group. DHA occurred in a higher proportion than EPA in all treatment groups. The sum of DHA

and EPA reached 28.9%, 34.6%, 32.7%, 30.5%, and 35.5% for HS, NHS, HK, NHK, and PF, respectively.

**Table 8.** Fatty acid composition (area% of total lipid) of whole fish body of Pacific bluefin tuna for 9 days.

| | Initial | Final (28 dph) | | | | | $p$ | | |
|---|---|---|---|---|---|---|---|---|---|
| | (19 dph) | HS | NHS | HK | NHK | PF | I | H | I × H |
| 14:0 | 3.1 ± 0.6 | 1.6 ± 0.1 | 1.4 ± 0.2 | 1.6 ± 0.5 | 1.6 ± 0.1 | 0.9 ± 0.2 | ns | ns | ns |
| 16:0 | 25.4 ± 1.2 | 18.0 ± 0.7 [b] | 18.6 ± 0.8 [b] | 19.8 ± 0.9 [b] | 18.8 ± 2.2 [b] | 23.7 ± 0.4 [a] | ns | ns | ns |
| 16:1n−7 | 3.2 ± 0.5 | 2.5 ± 0.1 [bc] | 3.4 ± 0.2 [ab] | 3.2 ± 0.5 [ab] | 4.1 ± 0.2 [a] | 1.8 ± 0.3 [c] | * | * | ns |
| 18:0 | 11.3 ± 2.1 | 11.0 ± 0.9 [a] | 6.2 ± 0.3 [c] | 7.5 ± 0.2 [bc] | 6.1 ± 0.5 [c] | 8.9 ± 0.5 [b] | * | * | * |
| 18:1n−9 | 8.1 ± 1.0 | 8.3 ± 0.2 [c] | 10.6 ± 0.4 [b] | 9.9 ± 0.2 [bc] | 13.0 ± 1.3 [a] | 9.3 ± 0.6 [bc] | * | * | ns |
| 18:1n−7 | 2.0 ± 0.2 | 2.2 ± 0.0 [c] | 3.0 ± 0.1 [b] | 2.8 ± 0.1 [bc] | 3.7 ± 0.4 [a] | 2.2 ± 0.1 [c] | * | * | ns |
| 18:2n−6 | 2.2 ± 0.2 | 1.9 ± 0.2 [c] | 2.6 ± 0.1 [b] | 3.9 ± 0.3 [a] | 3.9 ± 0.2 [a] | 1.9 ± 0.1 [c] | * | ns | * |
| 18:3n−3 | 0.4 ± 0.1 | 0.2 ± 0.0 | 0.3 ± 0.0 | 0.2 ± 0.0 | 0.3 ± 0.0 | 0.2 ± 0.0 | ns | * | ns |
| 18:4n−3 | 0.7 ± 0.1 | 0.8 ± 0.3 | 0.6 ± 0.0 | 0.6 ± 0.2 | 0.8 ± 0.1 | 0.3 ± 0.1 | ns | ns | ns |
| 20:1n−9 | 0.5 ± 0.1 | 1.8 ± 0.2 [a] | 1.8 ± 0.1 [a] | 1.2 ± 0.3 [b] | 1.5 ± 0.1 [ab] | 0.3 ± 0.0 [c] | * | ns | ns |
| 20:1n−11 | 0.4 ± 0.1 | 1.4 ± 0.1 [ab] | 1.5 ± 0.1 [a] | 0.9 ± 0.2 [c] | 1.2 ± 0.1 [bc] | 0.3 ± 0.0 [d] | * | * | ns |
| 20:2n−6 | 0.1 ± 0.0 | 0.2 ± 0.0 [b] | 0.2 ± 0.0 [a] | 0.1 ± 0.0 [c] | 0.1 ± 0.0 [c] | 0.1 ± 0.0 [c] | * | * | * |
| 20:4n−6 | 3.0 ± 0.4 | 2.7 ± 0.4 [b] | 3.3 ± 0.2 [b] | 4.4 ± 0.6 [a] | 3.5 ± 0.2 [ab] | 3.7 ± 0.1 [ab] | * | ns | * |
| 20:4n−3 | 0.3 ± 0.1 | 0.2 ± 0.0 [b] | 0.3 ± 0.0 [a] | 0.2 ± 0.0 [ab] | 0.2 ± 0.0 [ab] | 0.3 ± 0.0 [ab] | ns | * | ns |
| 20:5n−3 | 5.6 ± 1.0 | 7.1 ± 0.4 [b] | 8.6 ± 0.4 [ab] | 8.2 ± 0.3 [ab] | 9.4 ± 0.8 [a] | 5.0 ± 0.3 [c] | * | * | ns |
| 22:5n−6 | 0.9 ± 0.1 | 0.6 ± 0.1 [b] | 0.9 ± 0.1 [ab] | 0.8 ± 0.1 [ab] | 0.9 ± 0.0 [ab] | 1.1 ± 0.2 [a] | ns | * | ns |
| 22:6n−3 | 22.8 ± 3.5 | 21.8 ± 3.9 [ab] | 26.0 ± 2.2 [ab] | 24.5 ± 3.5 [ab] | 21.1 ± 2.0 [b] | 30.5 ± 0.2 [a] | ns | ns | ns |
| 22:6n−3/20:5n−3 | 4.2 ± 0.6 | 3.1 ± 0.5 [b] | 3.0 ± 0.2 [b] | 3.0 ± 0.5 [b] | 2.3 ± 0.1 [b] | 6.1 ± 0.4 [a] | ns | ns | ns |
| Others | 9.8 ± 1.5 | 16.3 ± 5.4 | 8.8 ± 4.3 | 8.3 ± 2.1 | 7.7 ± 2.0 | 9.3 ± 1.8 | ns | ns | ns |
| Σn−3 LC−PUFA [1] | 30.7 ± 4.5 | 30.7 ± 4.0 | 36.6 ± 2.5 | 34.6 ± 3.2 | 32.7 ± 2.9 | 37.4 ± 0.2 | ns | ns | ns |

[1] LC−PUFA, long chain poly unsaturated fatty acids; Σn−3 LC−PUFA: 18:4n−3, 20:4n−3, 20:5n−3, 22:5n−3, 22:6n−3. Values are presented as means of triplication. Values in a same row with different superscript letters are significantly different (Tukey's test, $p < 0.05$). I, ingredients (squid meal and krill meal); H, heated and non-heated treatment; ns, no significant difference (two-way ANOVA, $p > 0.05$); *, $p < 0.05$. HS, heated squid meal; NHS, non-heated squid meal; HK, heated krill meal; NHK, non-heated krill meal; PF, prey fish; Spangled emperor larvae, *Lethrinus nebulosus*.

## 4. Discussion

Kvåle et al. (2009) [36] reported that different feeding practices greatly affect fish performance, and the weaning period and method of introducing formulated diets are important. In the present study, based on the findings of Haga et al. (2010) [37], Cho et al. (2016) [38] and Cho et al. (2022) [39], we fed yolk-sac larvae of spangled emperor and a commercial diet on the first day, but the frequency of feeding yolk-sac larvae was gradually reduced over the subsequent two days due to acclimation to the formulated diet. According to Cho et al. (2016) [38], early larvae of PBT is able to weaned to formulated diets once they successfully accepted and ingested a suitable formulated diet. Similarly, in the present study, no significant differences were observed in fish growth and survival immediately after the weaning period, but we were able to observe significant differences in growth performance after changing their diet (Figures 1–3).

In the present study, there were significant differences in the contents of water-soluble proteins and free amino acids between the heated and non-heated diets (Tables 1 and 3). Considering an extruded pelleting process which is subjected to heat treatment, protein denaturation of the non-heated meal could be expected. To avoid protein denaturation of the non-heated meal, an extruder was not used to prepare the test diet in the present study. According to Cho et al. (2018) [28], in a study comparing the effects of heat treatment of ingredients on the growth performance of juvenile red sea bream *Pagrus major*, the growth of fish fed a non-heated diet was significantly better than that of fish fed a heated diet. Cho et al. attributed this difference in growth performance to the high water-soluble protein and free amino acid content in the feed. Watanabe (1982) [40] reported the incorporation of water-soluble proteins by pinocytosis in rectal epithelium cells in larval and juvenile teleosts. In the present study, it is suggested that the highly water-soluble protein contained in the non-heated diets has been effectively absorbed and assimilated in the early stages of

PBT. Furthermore, higher utilization of free amino acids was demonstrated in the larvae and juvenile of fish [41,42]. Current result showed that approximately 1.7 times higher gross free amino acid contained in the non-heated diets than the heated treatment diets, suggesting that they were utilized more effectively for growth.

Satoh (2005) [43] reported the in vitro digestibility of fish meal and raw fish by pepsin and trypsin from yellowtail *Seriola quinqueradiata*. Both pepsin and trypsin activity of fish fed raw fish were much higher than in those fed fish meal, and these enzymes worked better on protein from raw fish compared to that from fish meal. Furthermore, Seoka et al. (2010) [27] reported that the growth and survival (at 30 days post hatching) of PBT juvenile fed non-heated Toyama squid *Watasenia scintillans* meal-based diet were better than that fed a commercial diet. Yellowtail and tuna species feed on live fish in natural ecosystems; therefore, it is thought that the proteins of non-heated treatment without denaturation are highly digestible; therefore, using a non-heated protein source in an aquaculture setting is likely to be superior in terms of digestibility. It was reported that PBT juveniles exhibited higher gustatory responses to alanine, leucine, valine, methionine, isoleucine, and proline [44]. Here we demonstrated that the gross levels of these amino acids in non-heated diets (1.90 and 1.16 g/100 g) were higher than those in the heated diets (0.85 and 0.64 g/100 g). These results suggest that a diet containing high levels of water-soluble proteinous component such as free amino acids that are effective for fish growth can be produced without heating the ingredients and such a diet will promote the feed intake of PBT juveniles. Furthermore, Marubeni Nisshin Feed Co, Tokyo, Japan, developed a commercial formulated diet (Magokoro) based on enzyme-treated Chilean fish meal called BioCP from LANDES Co., Talcahuano, Chile, for PBT juveniles. Higher water solubility after enzyme treatment has been well documented for fish protein [45], and 60% of protein in BioCP accounts for soluble protein. However, the production of BioCP is expected to decline in the near future. When fish meal is in short supply, non-heated proteins can be more competitive depending on the availability of soluble proteins in the market.

The effects of feed ingredient on nutritional status were investigated in the present study. As a result, nutritional status indices, including growth performance, whole body composition, constitutional amino acid, and fatty acid altered. Fish fed krill meal-based diet (HK and NHK) had a positive WG, SGR, TGC, and survival. It was suggested that krill meal diet promotes digestive ability of yellowtail by regulating the digestive enzyme secretion in intestine [43].

Compared to squid meal, krill meal is considered to be a more effective ingredient for marine fish, including for the early life stage of PBT. The crude protein in krill contains 40% of the extractable nitrogen-containing components that are not contained in the protein itself [46]. Since the extractable component contains free amino acids and peptides and does not require decomposition by proteases, it might be easily absorbed. In the present study, the content of water-soluble proteins in the krill meal was not high. This could be because the extract component was dissolved and lost when it thawed. The essential amino acid contents, except tryptophan and valine, of the whole fish body were affected by the feed ingredients. This finding suggests that the amino acid balance of krill meal makes it more suitable than squid meal for PBT juveniles.

Krill is rich in astaxanthin, which has antioxidant activity, whereas squid meal is low in astaxanthin [47]. Although the content of astaxanthin was not measured in the present study, it is expected that the krill meal-based diet contains astaxanthin, which is considered to be effective in preventing the oxidation of long-chain polyunsaturated fatty acids. The fatty acid 18:1n−9 was identified as the major monounsaturated fatty acid (MUFA) and was significantly higher in the heated treatment (HS and HK) and PF groups. The most abundant saturated fatty acid (SAFA) in the present study was 16:0, which is considered to be a predominant source of potential metabolic energy in fish during growth [48], and it is the predominant SAFA in the main live feed source of cultured tuna [48,49]. The ratio of DHA/EPA in raw feed for PBT was reported to exceed 2.0 [50], and the DHA/EPA ratio of

PF used in the present study was 4.0. Further, Seoka et al. (2008) [51] reported that PBT juveniles fed yolk-sac larvae of striped beakfish *Oplegnathus fasciatus* with a DHA/EPA ratio of 3.6 show better growth performance than those fed a formulated diet with a DHA/EPA ratio of 1.7. It has also been considered that the dietary DHA/EPA ratio affects the growth performance of marine fish larvae and juveniles, and a ratio of at least 1.0 is appropriate for normal growth [52,53]. Although the DHA/EPA ratio of the test diet used in the present study exceeded 1.0, the growth in the test group was lower than that of the PF group (4.0).

Although the present study clearly demonstrated that non-heated meal is a suitable protein source for formulated diet for PBT juveniles, overall survival of the PBT fed the formulated diets was not very high (25.8–52.6%). Considering high mortality of hatchery-raised PBT juvenile is the one of the biggest bottleneck of mass production of the PBT juveniles, further improvement of the feed performance of the formulated diet for PBT is desired.

## 5. Conclusions

In conclusion, it is important to understand whether the selection of a feed protein source and its heat treatment would improve the nutritional status and thus growth of PBT juveniles. The present study showed that heat treatment of the protein source in feed adversely affected growth performance and survival of PBT juveniles. Furthermore, the influence of heat treatment was more remarkable in squid meal than in krill meal. Taken together, the results of the present study suggest that protein source as well as heat treatment may influence the nutritional status of cultured juvenile PBT, and this should be taken into consideration for the management of seed production.

**Author Contributions:** J.-H.C.: conceptualization, methodology, data curation, writing—original draft, and visualization. T.K.: conceptualization, methodology, investigation, validation. Y.H.: Conceptualization, supervision, project administration, funding acquisition, writing, review, and editing. Y.K.: Investigation, validation, resources. A.I.: Conceptualization, funding acquisition, project administration. S.S.: Conceptualization, funding acquisition, project administration. All authors have read and agreed to the published version of the manuscript.

**Funding:** This research was funded by the Maruha Nichiro Co. and Fisheries Agency Japan.

**Institutional Review Board Statement:** All activities related to animal ethical considerations, such as anesthesia, dissection, and euthanasia, were conducted according to Handling Rules for Animal Experiments, etc., Tokyo University of Marine Science and Technology (13 March 2020, TUMSAT Regulations No. 8) based on Basic Guidelines for Conducting Animal Experiments at Research Institutes, etc. (Ministry of Education, Culture, Sports, Science and Technology), Act on Welfare and Management of Animals (Act No. 105), Guidelines for the Proper Implementation of Animal Experiments (Science Council of Japan), and Guidelines on How to Dispose of Animals (Prime Minister's Office).

**Data Availability Statement:** Not applicable.

**Acknowledgments:** The expense of this work was partly supported part by a project on the development of production techniques for healthy PBT juveniles from Maruha Nichiro Co. and a grant for Technological Development for Selection and Secure Stock of Brood stock for Culture of Bluefin Tuna from the Fisheries Agency.

**Conflicts of Interest:** The authors have no declaration of interest.

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
