# Peer review of "Effects of Non-Heated and Heat Processed Krill and Squid Meal-Based Diet on Growth Performance and Biochemical Composition in Juvenile Pacific Bluefin Tuna Thunnus orientalis"

_fishes, doi:10.3390/fishes7020083_

Round 1

Reviewer 1 Report

Overall I think this study was very well conducted with appropriate statistics and analyses. The manuscript is well written and I believe delivers novel methods and results for this highly important species with applications to many other species. My biggest comment is that I think there should be more description of the feed manufacturing process. Since part of the goal was to not heat the two primary ingredients for two of the feeds, I assume the final pellets were not extruded? but what was the mixing and cold pressing protocols in order to develop the pellets? My other concern is the dwindling survival in all treatments? What was the rational for for only carrying out the experiment for 6 days after the co-feeding/weaning process during the first 3 days? Could the experiment have been carried out longer, or at least until survival in the reference (PF) treatment had stabilized. This would allow a better interpretation of the survival numbers and separated out the phase with significant, non-related, die-off of larvae who do not begin feeding or do not switch effectively to pelleted feed (there seems to always be some % of fish that just don't wean). Without that data, it is hard to interpret too much from the survival numbers to know what is normal and/or typically observed in this age-range. Please address these concerns.

Author Response

Dear Reviewer 1

Enclosed for your consideration is an original article that has been revised per the comments from the reviewer 1, entitled “Effects of non-heated and heat processed krill and squid meal-based diet on growth performance and biochemical composition in juvenile Pacific bluefin tuna Thunnus orientalis, for consideration for publication in Fishes.

We have responded to the comments from the reviewer 1 (PLEASE SEE THE ATTACHMENT) and we hope our responses are adequate for our paper to be accepted for publication. We appreciate your time and look forward to your response.

Reviewer 2 Report

The manuscript give data on the possibilities to replace fishmeal (FM) by other marine resources in diets for juvenile Pacific blue tuna but, from my point of view,  the short duration of experiment is a main constraint to interpret the obtained results.
Also an important question arises: It is sustainable the use of krill and squid?. It is important consider this, as the use of other marine resources could lead to similar problems related to the use of FM. I feel this question should addressed in the introduction.
From my point of view, information of table 2, 3 and 4 should be moved to results.
Although the experiment only lasts 10 days, a high mortality was evidenced in all formulated diets (>47%). This led to consider that tested diets were nor adequate to tuna juveniles. However this important point is not discussed and authors only mentioned that there were no diffrences in the three-day weaning period.
Authors must explain why decide to finish the experiemnt after 10 days. Maybe the high mortality was the main reason ?

Author Response

Dear Reviewer 2

Enclosed for your consideration is an original article that has been revised per the comments from the reviewer 2, entitled “Effects of non-heated and heat processed krill and squid meal-based diet on growth performance and biochemical composition in juvenile Pacific bluefin tuna Thunnus orientalis, for consideration for publication in Fishes.

We have responded to the comments from the reviewer 2 (PLEASE SEE THE ATTACHMENT) and we hope our responses are adequate for our paper to be accepted for publication. We appreciate your time and look forward to your response.

Round 2

Reviewer 1 Report

The authors have fully addressed the concerns raised in the review process and the manuscript is acceptable. 

This manuscript is a resubmission of an earlier submission. The following is a list of the peer review reports and author responses from that submission.

Round 1

Reviewer 1 Report

Overall I think this study was very well conducted with appropriate statistics and analyses. The manuscript is well written and I believe delivers novel methods and results for this highly important species with applications to many other species. My biggest comment is that I think there should be more description of the feed manufacturing process. Since part of the goal was to not heat the two primary ingredients for two of the feeds, I assume the final pellets were not extruded? but what was the mixing and cold pressing protocols in order to develop the pellets? My other concern is the dwindling survival in all treatments? What was the rational for for only carrying out the experiment for 6 days after the co-feeding/weaning process during the first 3 days? Could the experiment have been carried out longer, or at least until survival in the reference (PF) treatment had stabilized. This would allow a better interpretation of the survival numbers and separated out the phase with significant, non-related, die-off of larvae who do not begin feeding or do not switch effectively to pelleted feed (there seems to always be some % of fish that just don't wean). Without that data, it is hard to interpret too much from the survival numbers to know what is normal and/or typically observed in this age-range. Please address these concerns.

Reviewer 2 Report

The manuscript give interesting data on the possibilities to replace fishmeal (FM) by other marine resources in diets for juvenile Pacific blue tuna but the short duration of experiment is a main constraint to interpret the obtained results.

 A important question arises: It is sustainable the use of krill and squid?. It is important consider this, as the use of other marine resources could lead to similar problems related to the use of FM. I feel this question should addressed in the introduction.

From my point of view, information of table 2, 3 and 4 should be moved to results.

Although the experiment only lasts 10 days, a high mortality was evidenced in all formulated diets (>47%). This led to consider that tested diets were nor adequate to tuna  juveniles. However this important point is not discussed and authors only mentioned that there were no diffrences in the three-day weaning period.

Authors must explain why decide to finish the experiemnt after 10 days. Maybe the high mortality was the main reason ?

Reviewer 3 Report

The only comment is that the manuscript required Moderate English editing (language and style).